# Widespread occurrence of large molecular methylsiloxanes in ambient aerosols

Peng Yao<sup>1</sup>, Rupert Holzinger<sup>2,\*</sup>, Beatriz Sayuri Oyama<sup>3</sup>, Agne Masalaite<sup>4</sup>, Dipayan Paul<sup>1</sup>, Haiyan Ni<sup>1,5</sup>, Hanne Noto<sup>2</sup>, Dušan Materić<sup>2,6</sup>, Maria de Fátima Andrade<sup>3</sup>, Ru-Jin Huang<sup>5</sup>, Ulrike Dusek<sup>1,\*</sup>

- <sup>1</sup>Centre for Isotope Research (CIO), Energy and Sustainability Research Institute Groningen (ESRIG), University of Groningen, Groningen, 9747AG, the Netherlands

  <sup>2</sup>Institute for Marine and Atmospheric Research IMAIL Utracht University Princetonplain 5, 3584 CC. Utracht, the
  - <sup>2</sup>Institute for Marine and Atmospheric Research, IMAU, Utrecht University, Princetonplein 5, 3584 CC, Utrecht, the Netherlands
  - <sup>3</sup>Institute of Astronomy, Geophysics and Atmospheric Sciences, University of São Paulo, São Paulo, Brazil
- 4State Research Institute Center for Physical Sciences and Technology, Vilnius, LT-02300, Lithuania
   5State Key Laboratory of Loess and Quaternary Geology, Center for Excellence in Quaternary Science and Global Change, Key Laboratory of Aerosol Chemistry & Physics, Institute of Earth Environment, Chinese Academy of Sciences, Xi'an 710061, China
- <sup>6</sup>Department of Analytical Chemistry, Helmholtz Centre for Environmental Research UFZ, Permoserstrasse 15, 04318 5 Leipzig, Germany

Correspondence to: Rupert Holzinger (r.holzinger@uu.nl) and Ulrike Dusek (u.dusek@rug.nl)

Abstract. Synthetic pollutants have emerged as a widespread environmental concern. Recently, large molecular methylsiloxanes were identified in traffic emissions. Here, we show that large molecular methylsiloxanes are widely present in atmospheric particulate matter across diverse environments, including urban, coastal, rural, and forest sites in the Netherlands, Lithuania, and Brazil. Overall, methylsiloxanes of varying molecular sizes account for approximately 2.0%–4.3% of the non-refractory organic aerosol mass detected by thermal desorption proton transfer reaction mass spectrometry (TD-PTR-MS) analysis. Thermal desorption profiles indicate that over half of the detected methylsiloxanes originate from the depolymerization of large molecular methylsiloxanes, primarily associated with traffic emissions, while the remainder likely arise from the gas-to-particle conversion of volatile methylsiloxanes. Large molecular methylsiloxanes show a distinct correlation with long-chain hydrocarbons characteristic of engine lubricants, suggesting a lubricant-related source. Notably, the mass fraction of methylsiloxanes in organic aerosols does not decrease significantly during atmospheric transport and dilution, and a substantial fraction persists as large molecular methylsiloxanes. This persistence underscores their chemical stability, in contrast to the co-emitted lubricant hydrocarbons that undergo atmospheric oxidation. The substantial mass fraction of methylsiloxanes in particulate matter highlights their role as one of the most concentrated categories of synthetic compounds in the atmosphere, raising concerns about their potential, yet poorly understood, effects on human health and the climate.

#### 1 Introduction

Synthetic materials have improved human convenience, yet their pervasive presence in the environment raises concerns for ecosystems and human health. Recent attention has focused on certain synthetics, notably microplastics (Jambeck et al., 2015) and per- and polyfluoroalkyl substances (PFAS) (Evich et al., 2022), with global annual productions of 390.7 (PlasticsEurope, 2022) and 0.23 million tons (Glüge et al., 2020), respectively. Siloxanes, another class of synthetic compounds extensively used in/as lubricants, had a global annual production of approximately 3.48 million tons in 2023 (Global Silicones and Siloxanes Market Outlook). These synthetic organic compounds feature alternating silicon and oxygen arrangements in their skeletons, predominantly manifesting as methylsiloxanes (Rücker and Kümmerer, 2015). Despite the fact that some volatile species have been proved to be health hazards (United States Environmental Protection Agency (US EPA), 2022; Candidate List of substances of very high concern for Authorisation), such as affecting estrogen and liver (Quinn et al., 2006; Franzen et al., 2017; Gentry et al., 2017; Franzen et al., 2016), methylsiloxanes have received limited attention. Polydimethylsiloxanes (PDMS, or silicone oil; CH<sub>3</sub>[Si(CH<sub>3</sub>)<sub>2</sub>O]<sub>n</sub>Si(CH<sub>3</sub>)<sub>2</sub>) are commonly used in small amounts as defoaming agents in lubricants and can sometimes function directly as lubricants. Cyclic volatile methylsiloxanes (cVMS; ((Si(CH<sub>3</sub>)<sub>2</sub>O)<sub>n</sub>, n = 3–n), with their "roll-like-a-wheel" structure, can commonly serve as lubricants themselves. Given their extensive use, methylsiloxanes released into the atmosphere may have implications for both human health and climate, yet these potential impacts remain poorly studied.

Until recently, the main origins of methylsiloxanes detected in the atmosphere were attributed to the volatilization of small molecular methylsiloxanes in personal care and industrial products (Tang et al., 2015; Yucuis et al., 2013; Coggon et al., 2018).

Consequently, prior research predominantly focused on gas-phase volatile methylsiloxanes (VMS), while those in the particle phase were mainly considered to be byproducts of gas-to-particle transfer (Bzdek et al., 2014; Janechek et al., 2019; Han et al., 2022). However, our recent studies have revealed substantial quantities of particle-phase methylsiloxanes emitted from ships (Yao et al., 2022) and vehicles (Yao et al., 2023), indicating far greater emissions of methylsiloxanes than previously presumed. Notably, the methylsiloxanes emitted by ships and vehicles include a significant fraction of large molecular methylsiloxanes, which were detected by mass spectrometry after high-temperature thermal depolymerization into smaller molecules. This differs from previously identified small molecular VMS added in personal care and industrial products, representing a novel pollutant that has remained undetected. Therefore, the overall presence of methylsiloxanes in the atmospheric aerosols may have been significantly underestimated, resulting human inhalation and fate of these small and large molecular methylsiloxanes poorly understood.

Employing a high-resolution mass spectrometry analysis technique based on the characteristic peak distribution of natural silicon isotopes (Yao et al., 2022, 2023), our objective is to quantify methylsiloxanes in atmospheric particulate matter across diverse environments and identify their primary source. Particular attention is given to large molecular methylsiloxanes and their co-emission with long-chain hydrocarbons from lubricants, which may provide insights into atmospheric processes. We further explore the human inhalation and potential environmental impacts of large molecular methylsiloxanes.

90

#### 2. Materials and methods

#### 2.1 Sampling of atmospheric particulate matter

Particulate matter samples were collected from diverse environments. In the Netherlands, sampling was carried out at the Cesar Observatory (51.97° N, 4.90° E) (Dusek et al., 2017), a rural site situated in a heavily industrialized region in Western Europe, located between the Hague, Rotterdam, Amsterdam, and Utrecht. Particulate matter samples with sizes smaller than 2.5  $\mu$ m (PM<sub>2.5</sub>) were collected on pre-cleaned quartz filters (Whatman, QM-A; heat at 800 °C for12 h) using a high-volume sampler (Digitel DHA-80) and a flow rate of 500 L min<sup>-1</sup>. Samples were collected separately for day and night over a period of several days during different seasons (n = 38), specifically from 11 February 2011 to 16 October 2012, and the sampling duration varied by seasons, ranging from 2 to 9 days. Before and after sampling, the filters were kept wrapped in pre-backed aluminum foil and stored at -20 °C.

Aerosol samples were collected at three representative sites in Lithuania, i.e., Vilnius (urban, 54.63° N, 25.17° E), Preila (coastal, 55.37° N, 21.02° E), and Rugsteliskis (forest, 55.45° N, 26.00° E), representing typical northern European environments. Sampling campaigns were carried out during winter, when ambient temperatures averaged around –4 °C: from 30 December 2008 to 26 January 2009 in Vilnius, 8–15 December 2012 in Preila, and 1–29 March 2013 simultaneously at all three sites.

During the 2008–2009 and 2012 campaigns (Masalaite et al., 2018),  $PM_1$  aerosol samples were collected at Vilnius (n = 5) and Preila (n = 4) using a micro-orifice uniform deposition impactor (MOUDI-110) equipped with 11 stages and operated at a flow rate of 30 L min<sup>-1</sup>. Aluminum foils (47 mm diameter) were used as collection substrates and preheated at 600 °C for 10 h to remove trace organic contaminants prior to sampling. Although this temperature was relatively high for aluminum, no abnormal effects were observed in final results.

In March 2013, PM<sub>1</sub> aerosol samples were collected simultaneously at all three sites (Masalaite et al., 2017). At Vilnius (n = 19), samples were collected on 150 mm quartz microfiber filters (Whatman QM-A) using a high-volume sampler (Digitel DH-77) operating at 500 L min<sup>-1</sup>. At Preila (n = 6), a low-volume sampler (Leckel) operating at 30 L min<sup>-1</sup> was used with 47 mm quartz filters, while at Rugsteliskis (n = 6), samples were collected on 75 mm aluminum foils using a high-flow, three-stage impactor (MOUDI 128) operated at 100 L min<sup>-1</sup>. Filters and foils were preheated at 550 °C for 12 h prior to sampling to eliminate residual organics. After collection, the samples were wrapped in pre-fired aluminum foil (500 °C, 12 h), sealed in plastic bags, and stored at -25 °C until analysis.

In Brazil, particulate matter samples were collected in São Paulo (23.56° S, 46.73° W) (Oyama et al., 2016), a large metropolitan area in South America. Daily ambient samples were collected during the winter of Southern Hemisphere, specifically from 6 July to 9 September 2012, with an average temperature of approximately 18 °C. PM<sub>2.5</sub> samples (n =31) were collected on pre-cleaned quartz filters (Whatman, QM-A; heated at 800 °C for 12 h) using a high-volume sampler and a

flow rate of 1.13 m<sup>3</sup> min<sup>-1</sup>. Following the sampling procedure, the collected samples were immediately enveloped in precleaned aluminum foil (550 °C for 8h), sealed within polyethylene bags, and subsequently stored in low-temperature freezers at -18 °C until analysis.

Despite differences in sampling methods and particle size cutoffs, we do not expect these factors to substantially affect the derived methylsiloxane fractions. It should be noted that PM<sub>1</sub> typically accounts for 60–90% of the PM<sub>2.5</sub> mass (Theodosi et al., 2011; Liu et al., 2024); therefore, the methylsiloxane concentrations in Lithuanian PM<sub>1</sub> aerosols are expected to be approximately 10–40% lower than those in PM<sub>2.5</sub> samples from other regions.

#### 2.2 Chemical analysis

105

The chemical composition analysis of the collected particulate matter samples was conducted using a thermal desorption – proton transfer reaction - time of flight - mass spectrometer (TD-PTR-ToF-MS, PTR-TOF8000, Ionicon Analytik GmbH, Austria) (Holzinger et al., 2010b, a) with a mass resolution of 3000-4000 at full width at half maximum. The particulate matter samples were thermally desorbed from the filters using an oven with temperature steps of 3 min from 100 °C to 350 °C in 50 °C increments for the Lithuanian and Brazilian samples, and of 3 min form 125 °C to 350 °C in 75 °C increments for the samples collected in the Netherlands. A pure nitrogen carrier gas at a flow rate of 100 mL min<sup>-1</sup> or 50 mL min<sup>-1</sup> was used for carrying the desorption products to the PTR-MS. To prevent condensation of organic compounds, the drift tube and inlet line temperatures were maintained at 120 °C and 180 °C, respectively. A higher temperature could provide additional assurance against potential condensation losses, but increasing the drift tube temperature further is currently technically challenging for us. In this study, we focused solely on the desorbed organic aerosol fraction up to 350 °C, which is referred to as non-refractory organic aerosols (OA) in this study, and is not very sensitive to the individual steps in the heating protocol. We did not examine the non-desorbed fraction due to the limitation of our custom-designed heating unit. As indicated by our previous study, large molecular methylsiloxanes, such as PDMS with 10,000 cSt and 800 siloxane units, can withstand temperatures exceeding 650 °C (Yao et al., 2023). Consequently, some large molecular methylsiloxanes may not have undergone complete thermal decomposition in our experiments. The total organic aerosol concentration, including the non-desorbed fraction, would also be higher than the reported non-refractory OA values. Detailed discussion regarding thermal desorption and PTR-MS analysis of organics refers to our previous studies (Holzinger et al., 2010b; Timkovsky et al., 2015; Oyama et al., 2016; Materić et al., 2017).

PTRwid software was employed to identify and integrate peaks (Holzinger, 2015) in the mass spectra, to give the concentration of individual compounds in ppb in the carrier gas. The instrument was initially calibrated with standards to establish a transmission curve (Fig. S9), which as then applied to convert signal intensity into concentration (Holzinger et al., 2019; Worton et al., 2023). A unified mass list comprised the mass-to-charge ratio (m/z) with  $\pm$  2 standard deviations (95% confidence interval) for all identified peaks, along with potential molecular formulas having m/z within this range of uncertainty. The median values of the system blanks and the field blanks were subtracted from the mass spectra. Most samples

were analyzed in triplicate, with median concentration values used for further calculations. Detection limits corresponded to 1.5% of the sample concentrations, calculated as the mean of system blanks plus three times the standard deviation. The uncertainty is approximately 20.3% of the measured concentration, evaluated as the ratio of the standard deviation to the mean of replicate measurements. Due to the relatively large variability in sample concentrations, we assess uncertainty using this percentage-based approach. This level of uncertainty is slightly higher than that of conventional mass spectrometry methods but is typical for TD-PTR-MS (Holzinger et al., 2010b), as thermal desorption processes and protonation probabilities can introduce additional variability.

#### 2.3 Identification and quantification of methylsiloxanes

Upon heating of the filter samples, smaller methylsilxanes contained in the aerosol particles thermally desorb at relatively low temperatures. On the other hand, large molecular methylsiloxanes (e.g., PDMS) undergo depolymerization into small volatile methylsiloxanes (VMS) and fragments, with characteristic concentration ratios, dominated by D<sub>3</sub>, followed by D<sub>4</sub>, D<sub>5</sub>, and others (Yao et al., 2023; Thomas and Kendrick, 1969; Camino et al., 2001). The VMS resulting from both desorption and depolymerization were detected by the PTR-MS and identified based on the abundance of silicon isotopes, i.e.,  $^{28}$ Si (m/z = 27.977 amu, 92.223%),  $^{29}$ Si (m/z = 28.976 amu, 4.685%), and  $^{30}$ Si (m/z = 29.974 amu, 3.092%) (Yao et al., 2022, 2023). The isotope peaks of methylsiloxane molecules exhibit significantly higher intensities compared to other organic compounds with CHON structures, owing to the high abundance of  $^{29}$ Si and  $^{30}$ Si isotopes. For example, the main peak of D<sub>5</sub> (C<sub>10</sub>H<sub>30</sub>O<sub>5</sub>Si<sub>5</sub>) is observed at m/z = 371.102. The first isotope peak ranges from m/z = 372.101 to 372.106 and accounts for 36.41% of the main peak height, and the second isotope peak ranges from m/z = 373.099 to 373.108 and accounts for 23.56% of the main peak height. In contrast, organic molecules with CHON structures exhibit lower isotope peaks. For instance, the main peak of  $C_{11}H_{18}O_{12}N_2$  at m/z = 371.098 cannot be distinguished from  $D_5$  with the mass resolution of PTR-MS. However, the first isotope peak of  $C_{11}H_{18}O_{12}N_2$  only reaches 13.29% of the main peak height, and the second isotope peak only reaches 3.25%. With the characteristic ratios of the first and second isotope peaks to the main peak, various methylsiloxane depolymerization products and their derivatives were identified, including cVMS (D<sub>3</sub>-D<sub>10</sub>) and positively charged fragments, monomer fragments (CH<sub>3</sub>)<sub>2</sub>(OH)<sub>2</sub>Si and (CH<sub>3</sub>)<sub>2</sub>(OH)Si<sup>+</sup> (DMSD and DMHS<sup>+</sup>), and hydroxylated methylsiloxanes and related positively charged fragments. The quantification approach is based primarily on minimizing interference of other chemical compounds occurring either the same m/z as the main peak or as the isotope peaks (Yao et al., 2022, 2023). Details on the identification and quantification of methylsiloxanes are given in Method S1. The identified methylsiloxane molecules resulting from desorption and depolymerization of the large molecular methysiloxanes in each aerosol sample are shown in Figs. S1-S6 and Notes S1-S4.

160

140



#### 2.4 Estimation of human inhalation rate

The inhalation rate is determined by multiplying the inhalation volume by the pollutant concentration per unit volume. The typical 24-hour inhalation volumes for children and adults were sourced from previous studies (Allan and Richardson, 1998; Stifelman, 2007). Methylsiloxane concentrations in particulate matter were derived from the results of this study. Concentrations of PFAS were taken from established literature sources (Faust, 2023). See Method S2 for details.

Atmospheric microplastics are conventionally reported within a size range of  $1{\text -}5000~\mu\text{m}$  in diameter, focusing on particle number concentrations. To evaluate mass concentrations of inhalable microplastics ( $1{\text -}10~\mu\text{m}$ ), a custom model was developed based on a combination of assumptions and previously reported data related to microplastic particle number concentrations, morphological characteristics, size distribution, and density distribution. There is currently no consensus on whether nanoplastic concentrations in the sub-1  $\mu$ m size range are higher or lower than those of microplastics. Consequently, mass concentrations of nanoplastics up to 1  $\mu$ m in size were estimated by presuming that they exhibit similar PM<sub>1</sub> to PM<sub>10</sub> ratios as found in particulate matter. More details are available in Model S1.

#### 3 Results


# 3.1 Methylsiloxanes in atmospheric particulate matter

**Fig. 1. Spatial and temporal variations of methylsiloxanes in atmospheric particulate matter (PM).** Concentrations (**A**) and fractions (**B**) of methylsiloxanes in non-refractory organic aerosols (OA, thermally-desorbed up to 350 °C) at various locations: coastal (number of







samples n = 11, winter), urban (n = 23, winter), forest (n = 6, winter) in Lithuania (LTU; PM<sub>1</sub>); rural in the Netherlands (NL; n = 38, four seasons; PM<sub>2.5</sub>); and urban in São Paulo, Brazil (BR; n = 31, winter; PM<sub>2.5</sub>). Concentrations (**C**) and fractions (**D**) of methylsiloxanes in organic aerosols during different seasons in the Netherlands. (**E**) Processes occurring during sample heating for TD-PTR-MS analysis: (1) thermal desorption of small molecular methylsiloxanes and (2) thermal depolymerization of large molecular methylsiloxanes contained in the aerosol samples.

Although methylsiloxanes in atmospheric aerosols have been studied to some extent, knowledge about large molecular methylsiloxanes, compounds that only undergo thermal depolymerization at high temperatures, remains almost entirely lacking. To address this gap, we quantified particle-phase methylsiloxanes across a range of diverse environments, including coastal, urban, rural, and forest regions. Sampling locations was conducted in the Netherlands (West Europe), Lithuania (Northeastern Europe), and São Paulo in Brazil (South America). Concentrations of particle-phase methylsiloxanes and their mass fractions within non-refractory organic aerosols (OA) were determined in these diverse environments (Fig. 1A–1B).

Methylsiloxanes are widespread across diverse environments and occur at substantial concentrations (Fig. 1). It should be noted that the Lithuanian samples represent PM<sub>1</sub>, whereas the samples from the Netherlands and Brazil correspond to PM<sub>2.5</sub>. Since PM<sub>1</sub> typically accounts for 60–90% of PM<sub>2.5</sub> mass (Theodosi et al., 2011; Liu et al., 2024), the methylsiloxane concentrations in Lithuanian aerosols are expected to be 60–90% lower. The median concentrations of methylsiloxanes were highest in urban areas, 33 ng m<sup>-3</sup> in Lithuania (PM<sub>1</sub>) and 98 ng m<sup>-3</sup> in Brazil (PM<sub>2.5</sub>), and lowest in the forest site (0.9 ng m<sup>-3</sup> in Lithuania, PM<sub>1</sub>). Intermediate levels were observed in coastal (23 ng m<sup>-3</sup> in Lithuania, PM<sub>1</sub>) and rural environments (28 ng m<sup>-3</sup> in the Netherlands, PM<sub>2.5</sub>). Generally, locations with higher population densities exhibited higher concentrations of methylsiloxanes, whereas locations with lower population densities showed lower concentrations. This observation can be explained by higher traffic emissions in the urban environment or/and higher residential indoor emissions. The median mass fractions of methylsiloxanes in OA varied from 2.0% in rural areas of the Netherlands to 4.3% in urban areas of Brazil. These values surpass those observed in primary emissions from ships (1.2%) (Yao et al., 2022) and vehicles (1.1%) (Yao et al., 2023). This enrichment suggests either higher atmospheric stability of the methylsiloxanes than other primary aerosol matter and/or additional particulate methylsiloxane formation via additional sources or secondary processes.

Filter samples covering a whole seasonal cycle were analyzed in the Netherlands, allowing us to determine the seasonal variations of methylsiloxanes in atmospheric particulate matter (Fig. 1C–1D). The median concentrations of methylsiloxanes were lowest in summer ( $16 \text{ ng m}^{-3}$ ) and highest in winter ( $74 \text{ ng m}^{-3}$ ) in the rural Netherlands, indicating significant differences (1-way ANOVA, p = 0.01). The median fractions of methylsiloxanes in OA varied between 1.7% (spring) and 3.3% (winter), remaining relatively stable throughout the year. These results suggest that, although particle-phase methylsiloxanes exhibit seasonal variability, they represent a continuous and persistent environmental concern throughout the year.

The heating of filter samples for PTR-MS analysis (Fig. 1E) involves two main processes: (i) thermal desorption of small molecular methylsiloxanes and their oxidized products, and (ii) thermal depolymerization of large molecular methylsiloxanes. Accordingly, the total particle-phase methylsiloxanes include contributions from both components, which will be further discussed later. VMS are primarily distributed in the gas phase due to their relatively high vapor pressures, partitioning mainly to the gas phase rather than the particle phase. Oxidation lowers vapor pressures of VMS and promotes gas-to-particle



partitioning; however, even oxidized VMS remain small molecules that volatilize upon mild heating. During thermal desorption, these small molecular methylsiloxanes and the associated oxidized products evaporate at relatively low temperatures from the aerosol filter sample. At high temperatures (e.g., > 200 °C), large molecular methylsiloxanes (e.g., PDMS) undergo depolymerization, yielding smaller cVMS ( $D_3$ – $D_n$ ) (Yao et al., 2023; Thomas and Kendrick, 1969; Camino et al., 2001) and other by-products. This thermal depolymerization was evident in our data, as small molecular cVMS are too volatile to persist in the particle phase at such high temperatures. Thus, the high-temperature signal reflects large molecular methylsiloxanes, which differ fundamentally from gas-to-particle VMS both in properties and in sources.

# 3.2 Atmospheric methylsiloxanes and traffic emissions

Fig. 2. Characteristics of methylsiloxanes detected from heating traffic and atmospheric particulate matter samples at various desorption temperatures. Detected methylsiloxanes include desorbed molecules as well as depolymerization products of large molecular methylsiloxanes. Concentrations of various methylsiloxanes as a function of desorption temperature in (A) tunnel and (B) urban particulate








matter in São Paulo, Brazil. The relative mass fraction of methylsiloxanes released at temperatures  $\leq 200$  °C and 200-350 °C for (C) particulate matter sampled in two tunnels (vehicle) and from the smoke stack of a high-speed passenger hydrofoil (ship); (D) particulate matter sampled at coastal, urban, and forest sites in Lithuania; (E) particulate matter sampled at a rural site in the Netherlands across various seasons; and (F) particulate matter sampled in urban São Paulo, Brazil.

After initially detecting large molecular methylsiloxanes in traffic-related source samples (Yao et al., 2023), we now confirm their widespread and substantial presence in atmospheric aerosols from three countries and four diverse environments. Methylsiloxanes detected at high desorption temperatures (e.g., > 200 °C) can be attributed almost entirely to large molecular methylsiloxanes, with traffic currently recognized as their only known source. In contrast, the low-temperature fractions contain both smaller fractions of large molecular methylsiloxanes (depolymerized  $\le 200$  °C) and a considerable contribution from VMS that likely entered the particle phase through gas-to-particle conversion. These VMS can originate from household and industrial emissions and partition into aerosols via oxidation, adsorption, and related processes.

To preliminarily distinguish these sources in aerosols, we compared the concentrations and relative mass fractions of methylsiloxanes detected at different desorption temperatures steps in traffic sources (vehicle and ship) and ambient aerosol samples (Fig. 2). As outlined in the thermal desorption and depolymerization mechanism (Fig. 1E), the species observed across all temperature steps were mainly smaller cVMS (D<sub>3</sub>–D<sub>n</sub>, Fig. 2A–2B) and their characteristic fragments resulting from the loss of a –CH<sub>3</sub> group (Yao et al., 2022, 2023; Worton et al., 2023). We also identified dimethylsilanediol (DMSD, (CH<sub>3</sub>)<sub>2</sub>(OH)<sub>2</sub>Si) and its associated fragment dimethylhydroxysilyl cation (DMHS<sup>+</sup>, (CH<sub>3</sub>)<sub>2</sub>(OH)Si<sup>+</sup>), consistent with monomeric diols produced via hydrolysis of methylsiloxanes. Small amounts of hydroxylated methylsiloxanes were detected as well, and their concentrations were grouped with the major cVMS (see Method S1 for details).

A direct comparison of source and ambient aerosols from the same region (São Paulo, Brazil) illustrates how traffic emitted large molecular methylsiloxanes mix with gas-to-particle VMS in the atmosphere. In tunnel samples, methylsiloxane concentrations varied systematically with desorption temperature, and similar trends were evident in ambient aerosols (Figs. 2A-2B, fractions in Fig. S6). Previous studies have shown that gaseous VMS typically follow the distribution  $D_5 > D_4 > D_3$  (Yucuis et al., 2013; Genualdi et al., 2011; Brunet et al., 2024). However, PDMS depolymerization usually yields predominantly  $D_3$ , followed by  $D_4$ ,  $D_5$ , and others ( $D_3 > D_4 > D_5$ ; (Yao et al., 2023; Thomas and Kendrick, 1969; Camino et al., 2001)). These contrasting trends were evident across different temperature steps during thermal analysis. In both tunnel and ambient aerosols,  $D_5$  was the most abundant methylsiloxane at a desorption temperature of 100 °C, consistent with gas-to-particle conversion. Above 150 °C,  $D_3$  becomes the most abundant methylsiloxane, highlighting contributions from depolymerization of large molecular methylsiloxanes. At 150 °C, the concentrations of  $D_8-D_{10}$  increased compared to 100 °C, suggesting an additional contribution from gas-to-particle conversion, where larger molecules ( $D_8-D_{10}$ ) desorb at slightly higher temperatures than smaller molecules ( $D_5-D_7$ ). In contrast, methylsiloxane fractions desorbed above 200 °C were dominated almost exclusively by large molecular methylsiloxanes, confirming that these high-temperature signals represent a new class of pollutants. Overall, tunnel source samples were characterized primarily by large molecular methylsiloxanes with minor contributions from gas-to-particle conversion. In contrast, ambient aerosols in the same region contained strong




signatures from both sources, with large molecular methylsiloxanes contributing slightly more than VMS originating from gas-to-particle conversion.

Because thermal desorption signatures of gas-to-particle VMS have not yet been systematically characterized and no reliable source profiles exist, a full source apportionment of aerosol-phase methylsiloxanes is not currently possible. Nevertheless, since  $\leq 200$  °C fractions are contributed by both gas-to-particle VMS and large molecular methylsiloxanes, whereas 200–350 °C fractions are composed almost exclusively of large molecular methylsiloxanes, this temperature threshold provides a practical way to broadly assess their relative contributions across different regions. The relative mass fractions of methylsiloxanes detected at lower ( $\leq 200$  °C, both gas-to-particle VMS and large molecular methylsiloxanes) versus higher temperatures (200–350 °C, predominantly large molecular methylsiloxanes) were compared for samples from traffic sources and ambient atmospheric aerosols (Figs. 2C–2F). The depolymerization of large molecular methylsiloxanes is temperature-dependent, with larger molecules requiring higher temperatures to break down. Methylsiloxanes from ship emissions (Fig. 2C) exhibited depolymerization predominantly at 200–350 °C, whereas methylsiloxanes from vehicle emissions predominantly at  $\leq 200$  °C. These results imply that ships emit a higher proportion of larger methylsiloxane molecules than vehicles, attributed to differences in engine design and lubrication (Yao et al., 2023).

In Lithuania, the mass fractions of methylsiloxanes at 200–350 °C were higher at a coastal site downwind of a harbor than in urban locations (Fig. 2D), consistent with a higher contribution of ship emissions at the coastal location and a higher contribution of vehicle emissions in the urban region. Dutch samples (Fig. 2E) were collected at a rural background station, located between major harbor cities (Rotterdam, Amsterdam, and The Hague) and inland cities (Utrecht and Eindhoven). At this site, mass fractions of methylsiloxanes at 200–350 °C were lower in winter compared to other seasons. Air mass back trajectory analysis (Fig. S10) revealed that winter air masses were primarily continental, whereas in other seasons they arrived from the west over the ocean. This seasonal contrast indicates that ship emissions substantially increase the high-temperature fraction outside of winter. In São Paulo, Brazil (Fig. 2F), the thermal desorption profiles of urban aerosols closely resembled those of vehicle source samples collected in two local tunnels (Fig. 2C). The strong similarity in desorption characteristics, with minor differences likely due to gas-to-particle transformation, indicates that vehicle emissions are the dominant source of methylsiloxanes in São Paulo's atmosphere.

Overall, large molecular methylsiloxanes represent a substantial fraction of atmospheric aerosols in both Lithuania and the Netherlands, where 200–350 °C fractions already accounted for more than half of the total. This highlights that these newly identified large molecular methylsiloxanes are far more abundant than previously recognized gas-to-particle VMS. In São Paulo, the 200–350 °C fraction contributed more than one-quarter of the total, and depolymerization signatures of large molecular methylsiloxanes were already apparent at 150 and 200 °C, together accounting for over half of the total.



#### 3.3 Atmospheric transport, dilution, and oxidation

Fig. 3. Scatter plot of methylsiloxanes versus C<sub>23</sub>–C<sub>38</sub> hydrocarbons in tunnel and atmospheric particulate matter. The tunnel particulate matter was collected in São Paulo (Brazil, BR), while ambient samples include atmospheric particulate matter collected in São Paulo, Lithuania (LTU), and the Netherlands (NL).

Our results demonstrate that large molecular methylsiloxanes are widely present in atmospheric aerosols across diverse environments. Since traffic emissions are the only currently known large-scale source, an important question is how these compounds disperse into remote regions such as rural or forested areas. During atmospheric transport, dilution inevitably occurs, while additional methylsiloxanes may enter the particle phase via gas-to-particle conversion of VMS, and oxidation processes may further alter their composition. However, the relative importance of these processes remains uncertain.

In our tunnel study (Yao et al., 2023), we identified large molecular methylsiloxanes and long-chain hydrocarbons ( $C_{23}$ – $C_{38}$ ) from lubricants as co-emitted components of traffic particulate matter. These long-chain hydrocarbons are distinct markers of traffic emissions because they differ from shorter-chain hydrocarbons in gasoline ( $C_5$ – $C_{12}$ ), diesel ( $C_{12}$ – $C_{20}$ ), or biogenic sources. Previous studies reported their origins from internal combustion engines in vehicles, ships, and aircraft (Eichler et al., 2017; Masalaite et al., 2018; Yao et al., 2023; Decker et al., 2024; Sonntag et al., 2012). The co-occurrence of large molecular methylsiloxanes and long-chain hydrocarbons provides a tracer framework to track the transport and transformation of traffic-derived large molecular methylsiloxanes.

A comparison of tunnel and ambient samples in São Paulo illustrates how these tracers evolve in the atmosphere (Fig. 3). While both methylsiloxanes and long-chain hydrocarbons were strongly correlated in tunnel samples ( $R^2 = 0.83$ , Fig. S11), at the urban background site long-chain hydrocarbons decreased sharply, whereas methylsiloxanes declined much less. This divergence suggests faster atmospheric oxidation of long-chain hydrocarbons and greater persistence of large molecular methylsiloxanes, consistent with their higher chemical stability. Additional methylsiloxanes at the background site may also



originate from gas-to-particle transformation (Bzdek et al., 2014; Janechek et al., 2019; Han et al., 2022). Although methylsiloxanes can undergo oxidation by OH and Cl radicals (Alton and Browne, 2020), our isotopic analysis captures their oxidation and hydrolysis products, indicating that most remain accounted for in the particle phase.

In Lithuania, atmospheric particulate matter was collected during a cold winter (average -4 °C). Despite the vastly different environments, methylsiloxanes and long-chain hydrocarbons showed similar correlations at urban, coastal, and forest sites, comparable to those observed in São Paulo tunnels. In contrast to Sao Paulo, the two compound classes decreased at comparable rates from polluted to clean sites, pointing to dilution as the dominant atmospheric process. The cold temperatures and weak solar radiation likely suppressed oxidation, allowing long-chain hydrocarbons to persist longer in ambient aerosols. At the rural background site in the Netherlands, methylsiloxanes levels were of the same order of magnitude (or slightly lower) than in Lithuania, while long-chain hydrocarbons reached their lowest concentrations across all sites, suggesting stronger oxidative loss under milder climatic conditions.

### 4 Discussion and implications

#### 4.1 Methylsiloxanes inhalation

Fig. 4. Inhalation rates of methylsiloxanes in atmospheric particulate matter compared to other emerging pollutants. Estimated inhalation rates of (A) methylsiloxanes, (B) per- and polyfluoroalkyl substances (PFAS), and (C) micro- and nanoplastics (MNPs) in atmospheric particulate matter. Refer to Model S1 for details.

The potential health impacts of large molecular methylsiloxanes remain largely unknown and call for further investigation. Here, we provide a preliminary assessment of human exposure through inhalation, estimating the intake of aerosol-bound methylsiloxanes across different environments as a basis for future toxicological studies. Inhalation of atmospheric particulate






matter, especially fine particles (< 2.5 µm), can lead to deposition in the lungs. Although both short- and long-term health risks of such exposure remain uncertain, daily human intake can be estimated.

Our results show that inhalation rates of particle-phase methylsiloxanes vary substantially across environments (Fig. 4A). In Lithuania's forested areas, median daily intakes were 11.3 ng capita<sup>-1</sup> day<sup>-1</sup> for children and 13.0 ng capita<sup>-1</sup> day<sup>-1</sup> for adults, while in urban Brazil, values reached 1290 and 1480 ng capita<sup>-1</sup> day<sup>-1</sup>, respectively. Given the strong association of large molecular methylsiloxanes with traffic emissions, these results suggest that human exposure is a widespread and potentially significant issue on a global scale.

To contextualize our findings, we compared methylsiloxane inhalation with two other synthetic pollutants commonly reported in atmospheric particulate matter: PFAS and micro-/nanoplastics. Using literature-based aerosol concentrations of PFAS (Faust, 2023), we estimated median inhalation rates of 0.467 ng capita<sup>-1</sup> day<sup>-1</sup> for children and 0.550 ng capita<sup>-1</sup> day<sup>-1</sup> for adults (Fig. 4B), which are three to four orders of magnitude lower than the methylsiloxane intakes in urban and coastal regions. For micro- and nanoplastics, we converted particle number concentrations into mass concentrations by accounting for particle size distributions, morphology, and density (details in Model S1). Monte Carlo simulations were then applied, combined with literature-derived ranges of particle abundance (Mohamed Nor et al., 2021; Revell et al., 2021), yielding median inhalation estimates of 10.3 ng capita<sup>-1</sup> day<sup>-1</sup> for children and 12.1 ng capita<sup>-1</sup> day<sup>-1</sup> for adults (Fig. 4C). These values are one to two orders of magnitude lower than the methylsiloxane levels in urban environments. These comparisons highlight that, while the toxicological properties of methylsiloxanes are far less understood than those of PFAS or micro-/nanoplastics, the levels of human intake through inhalation can be substantially higher. This underscores the importance of investigating their potential health risks and long-term environmental impacts.

#### 4.2 Potential impacts of methylsiloxanes on aerosol properties

Methylsiloxanes are widely used in industrial formulations, notably as defoaming agents and anti-freeze components, where even small additions can alter the physicochemical properties of lubricants. In ambient aerosols from diverse environments, we detected substantial levels of large molecular methylsiloxanes, with total methylsiloxanes contributing 2.0–4.3% of organic aerosol and the large molecular fraction comprising more than half. These findings indicate that large molecular methylsiloxanes may exert an appreciable influence on the physicochemical properties of atmospheric aerosols.

One potential impact is through modification of aerosol surface tension. While atmospheric models typically assume a surface tension of 72 mN m<sup>-1</sup>, corresponding to pure water, PDMS can exhibit much lower values, down to 20 mN m<sup>-1</sup> (Gaines, 1969; Kanellopoulos and Owen, 1971; Ananthapadmanabhan et al., 1990), similar to those of traffic-related lubricant oils (~25 mN m<sup>-1</sup>) (Goebel and Lunkenheimer, 1997; Winoto et al., 2014). The interfacial tension between methylsiloxanes and water (30–50 mN m<sup>-1</sup>) (Kanellopoulos and Owen, 1971; El-Hamouz, 2007; Nowak et al., 2016) further supports the expectation that large molecular species may readily spread across droplet surfaces (Model S2 and Equation S5) (Harkins, 1952; Winoto et al., 2014; Soloviev et al., 2016). Such spreading could substantially alter the interfacial properties of aerosols. Simple Köhler





theory estimates (Equation S6) (Petters and Kreidenweis, 2007) suggest that, if sufficient large molecular methylsiloxanes coat aerosol surfaces, the critical supersaturation required for cloud droplet activation could be reduced by more than an order of magnitude. This magnitude of change is consistent with their industrial role as efficient defoaming agents, where trace additions strongly perturb fluid properties. However, direct experimental evidence for these atmospheric effects is currently lacking and should be pursued in future work.

Additionally, owing to the low freezing points, methylsiloxanes widely serve as constituents in anti-freeze formulations for the chemical industry and for cold-weather operations of vehicles and industrial equipment. Therefore, the presence of large molecular methylsiloxanes in atmospheric aerosols can potentially inhibit ice nucleation (IN), necessitating further investigation.

## 4.3 Implications

Fig. 5. Implications of large molecular methylsiloxanes in atmospheric particulate matter. (A) Health; (B) Atmospheric processes; (C) Indicator.

This study provides the first evidence that large molecular methylsiloxanes are widespread in atmospheric aerosols. As a newly recognized class of pollutants, their potential impacts on human health and the environment remain largely unknown, calling for urgent attention from both the scientific community and society at large. Substantial knowledge gaps exist, and systematic research is needed to address them. Overall, the potential environmental implications of methylsiloxanes in atmospheric particulate matter can be summarized in three main aspects (Fig. 5).

First, the inhalation of methylsiloxanes via particulate matter may pose potential health risks that require assessment. Second, the atmospheric fate of methylsiloxanes in particulate matter is not yet understood, including the processes of oxidation, fragmentation, hydrolysis, and other reactions occurring in both the gas and particle phase, as well as their impact on reactions of the organic and aqueous phases in aerosols. In addition, methylsiloxanes may alter the physical properties of aerosols, such as reducing surface tension (affecting their ability to act as cloud condensation nuclei) or exhibiting anti-freezing behavior (impacting ice nucleation). These possibilities also warrant further study. Third, methylsiloxanes are purely synthetic, relatively stable in atmospheric particulate matter, and emitted on a large scale, rendering them valuable indicators of

anthropogenic emissions and atmospheric transport processes, especially in remote regions. Collectively, these considerations

highlight the need for comprehensive and timely research on methylsiloxanes in atmospheric particulate matter, given their potential implications for human health, climate change, and environmental sustainability.

#### 5. Conclusion





Our measurements reveal that large molecular methylsiloxanes are abundant in atmospheric aerosols from three countries representing diverse environments. Methylsiloxanes contribute up to 2.0–4.3% of the total organic aerosol mass detected by thermal-desorption proton transfer reaction mass spectrometry using a maximum desorption temperature of 350 °C. Large molecular methylsiloxanes are detected via their characteristic fragmentation properties upon heating. The thermal-desorption profiles indicate that traffic emissions constitute a major source of large molecular methylsiloxanes across all ambient environments. Comparisons between tunnel and ambient air samples in Sao Paulo, Brazil, show that ambient air samples contain an increased fraction of methylsiloxanes of smaller molecular size at low desorption temperatures, presumably from gas-to-particle conversion processes. In the tunnel samples, methylsiloxanes and lubricating-oil hydrocarbon concentrations are highly correlated, indicating co-emission. During winter in Lithuania, methylsiloxanes and lubricating-oil hydrocarbons show a similar correlation in the ambient atmosphere, suggesting a common origin for both species and dilution as the major process during atmospheric transport. In Sao Paulo, methylsiloxane concentrations are similar in the tunnel and ambient atmosphere, whereas the concentration of lubricating-oil hydrocarbons decreases strongly in the ambient atmosphere. This shows the methylsiloxanes are chemically very stable and are likely to be transported over long distances. The estimated daily inhalation dose of methylsiloxanes via aerosols may exceed that of other synthetic compounds such as PFAS and MNPs, and studies assessing their toxicity are warranted. Their surface-tension-lowering and antifreezing properties may further influence the physical behavior and climatic effects of aerosols. Together, these findings highlight the ubiquity of this emerging pollutant and underscore the urgent need for systematic evaluation of its environmental and human health impacts.

#### Acknowledgments

This study was funded by the Netherlands Organization for Scientific Research (NWO, grants Nr. 820.01.001, and 834.08.002) and Foundation for Research Support of the São Paulo State (FAPESP; projects 2011/17754-2 and 2012/21456-0). P.Y. would like to express appreciation for the support from China Scholarships Council (grant No. 201806320346).

#### 420 Author contributions

Conceptualization: PY, RH, UD. Methodology: PY, RH, UD. Investigation: PY, BSO, AM, HNoto. Visualization: PY, RH, UD. Supervision: RH, MFA, UD. Writing—original draft: PY, RH, UD. Writing—review & editing: PY, RH, BSO, AM, H. Noto, DP, HNi, HNoto, DM, MFA, RJH, UD.

## **Competing interests**

The authors declare that they have no known competing financial interests or personal relationships that could have appeared to influence the work reported in this manuscript.

#### **Supporting Information**

Supporting Information for this study is available.

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
