# Peer review of "Widespread occurrence of large molecular methylsiloxanes in ambient aerosols"

_EGUsphere, 2025_

## Author Comment (AC1)

**Response to reviewer**

We thank the reviewer for the time and efforts spent on our manuscript and particularly for the valuable suggestions and comments that helped us improve the manuscript. We provide below point-by-point responses to the reviewers' comments and indicate how we implemented the changes suggested by the reviewers in the revised manuscript (blue text), with the reviewer' original comments in ***italic and bold***.
* * *
***This manuscript (Manuscript ID: egusphere-2025-5655) presents an original and timely investigation into the widespread occurrence of large molecular methylsiloxanes in ambient aerosols. The topic is both novel and important, as it highlights a previously underrecognized group of synthetic organic compounds within atmospheric particles. The multidisciplinary approach, combining measurements from diverse environments with detailed chemical characterization, is a significant strength of the study. The results are convincing and suggest that these compounds may play a more substantial role in atmospheric chemistry than previously appreciated. Overall, the work represents a valuable contribution to atmospheric chemistry and aerosol science, and I recommend publication after minor revision.***

***Here are some comments in details:***

***The term "large molecular methylsiloxanes" is central to the study but is currently used somewhat broadly. Please provide a clearer operational definition in the Methods section, including typical molecular weight ranges and the key diagnostic ions used for mass spectrometric identification.***

**Response:** We have expanded the Methods section to provide a clearer operational definition of "large molecular methylsiloxanes". Based on prior thermal tests of PDMS with different molecular weights (Yao et al., 2023), PDMS with a viscosity of 10 cSt, dominated by molecules containing approximately 15 siloxane units (with characteristic ions at $m/z = 1110$ for cyclic species), exhibits its main desorption/decomposition peak slightly below 200 °C. In contrast, higher-molecular-weight PDMS (e.g., 20 cSt, ~25 siloxane units) shows its dominant peak above 200 °C. Moreover, methylsiloxanes containing ~15 siloxane units have extremely low volatility under ambient conditions. Therefore, we

suggest ~15 siloxane units and 200 °C as practical thresholds for defining large molecular methylsiloxanes. This definition is now explicitly stated in the Methods section.

"Based on prior thermal tests of PDMS with different molecular weights (Yao et al., 2023), PDMS with a viscosity of 10 cSt, dominated by molecules containing approximately 15 siloxane units (with characteristic ions at m/z = 1110 for cyclic species), exhibits its main desorption/decomposition peak slightly below 200 °C. In contrast, higher-molecular-weight PDMS (e.g., 20 cSt, ~25 siloxane units) shows its dominant peak above 200 °C. Moreover, methylsiloxanes containing ~15 siloxane units have extremely low volatility under ambient conditions. Therefore, we suggest ~15 siloxane units and 200 °C as practical thresholds for defining large molecular methylsiloxanes." (Page 4, Line 116–121)

*While the study benefits from data collected at multiple sampling locations, it would be helpful to comment briefly on how representative these sites are of broader regional or global conditions. A short discussion of potential sampling biases (e.g., proximity to emission sources, meteorological influences) and the expected variability of methylsiloxane mass fractions across seasons and regions would strengthen the manuscript.*

**Response:** We have expanded the Discussion section to comment on the representativeness of the sampling sites and potential sampling biases. The samples analyzed in this study primarily originate from Europe and South America, and therefore may not fully capture the variability of methylsiloxane mass fractions in other regions such as Asia, Africa, or North America, which require further investigation. In addition, Lithuania has a relatively small geographic extent, with urban, coastal, and forested areas located in close proximity, which may lead to overlapping influences from multiple emission sources and meteorological transport. These limitations and their implications for regional and seasonal variability are now briefly discussed in the revised manuscript.

"It should be noted that the samples analyzed here primarily originate from Europe and South America, and may not capture methylsiloxane variability in regions such as Asia, Africa, or North America. Additionally, Lithuania's small geographic size, with urban, coastal, and forested areas in close proximity, may result in overlapping influences from multiple emission sources and local meteorology." (Page 15, Line 392–395)

*Line 21: The manuscript states that methylsiloxanes constitute 2.0–4.3% of the "non-refractory organic aerosol mass". Please clarify that this percentage refers specifically to the non-refractory organic component and acknowledge that refractory or less*

*volatile compounds are not included. A brief explanation would improve the transparency of the interpretation.*

**Response:** We thank the reviewer for this suggestion. The Methods section already addressed this issue, and we have now added additional details for clarity.

"In addition, we focused on the desorbed organic aerosol fraction up to 350 °C, hereafter referred to as non-refractory organic aerosols (OA). We did not investigate the non-desorbed fraction due to the technical limitation of our custom-designed heating unit. Previous work has shown that large molecular methylsiloxanes, such as PDMS with 10,000 cSt and 800 siloxane units, can withstand temperatures exceeding 650 °C (Yao et al., 2023). Consequently, part of the large molecular methylsiloxanes may not have undergone complete thermal decomposition in our experimental conditions, and both methylsiloxane and total organic aerosol concentrations reported here should be considered lower-bound estimates." (Page 4, Line 122–127)

*Line 70: "the Hague" should be corrected to "The Hague".*

**Response:** Corrected.

*Line 74: "pre-backed" should be revised to "pre-baked".*

**Response:** Corrected.

*Line 139: "methylsilxanes" should be corrected to "methylsiloxanes".*

**Response:** Corrected.

*Line 155: Please add the preposition "at" after "occurring".*

**Response:** Revised.

*Line 187: Replace "was" with "were".*

**Response:** Corrected.

*Line 192: The phrase "60–90% lower" appears inconsistent with the numbers presented and should be revised to "10–40% lower".*

**Response:** Revised.

*Line 234: The statement "traffic currently recognized as their only known source" is overly absolute. Please rephrase to acknowledge the possibility of additional or emerging sources.*

**Response:** Revised.

"with traffic currently recognized as their only known source (additional sources cannot be excluded)" (Page 10, Line 240)

*Line 345: The phrase "three to four orders of magnitude lower than the methylsiloxane intakes in urban and coastal regions" risks overgeneralizing all urban and coastal environments. Consider qualifying this comparison more carefully.*

**Response:** Revised.

"in urban and coastal regions reported in this study" (Page 14, Line 352)

*Line 411: Replace "may exceed that of other synthetic compounds" with "may be comparable to or exceed that of other synthetic compounds" for greater accuracy.*

**Response:** Revised.

---

## Author Comment (AC2)

**Response to reviewer**

We thank the reviewer for the time and efforts spent on our manuscript and particularly for the valuable suggestions and comments that helped us improve the manuscript. We provide below point-by-point responses to the reviewer's comments and indicate how we implemented the changes suggested by the reviewers in the revised manuscript (blue text), with the reviewer' original comments in ***italic and bold***.
* * *
***Review report of "Widespread occurrence of large molecular methylsiloxanes in ambient aerosols"***

***Methylsiloxanes are considered as an emerging class of pollutants. The authors have developed a novel analytical method called TD-PTR-TOF-MS, which enables to identify methylsiloxanes in the aerosol samples, and showing their widespread presence in ambient PM samples collected at diverse seasons and locations, accounting for 2-4% of organic aerosol mass fraction. The possible sources, correlation with long chain HC and atmospheric stability of methylsiloxanes are also discussed. In the end, the authors call for further attention to their potential health and climate impact.***

***I like this analytical idea based on Si isotope patten in combination with thermal desorption, which is highly selective and is actually a type of non-targeted strategy to identify Si-containing compounds in complicated samples, without using commercial standards. I strongly recommend it for publication and only have a few questions for the authors to address.***

***1. Can this analytical method be used to identify gas-phase methylsiloxanes? Also, it seems methylsiloxanes or cyclic volatile methylsiloxanes are sub-class of Si-containing compounds. Could other sub-class of Si-containing compounds (not methylsiloxanes or cyclic volatile methylsiloxanes) exist in the PM samples and be detected by TD-PTR-TOF-MS?***

**Response:** Yes, this analytical approach can also be applied to the identification of gas-phase methylsiloxanes, if their concentrations are above the detection level of the PTR-MS. In fact, some of the PTR-MS calibration compounds are gas-phase methylsiloxanes. In principle, other subclasses of silicon-containing compounds, beyond methylsiloxanes and cyclic volatile methylsiloxanes, could also

be detected by TD-PTR-TOF-MS if present and ionizable under the applied conditions. However, in the aerosol samples analyzed in this study, we did not observe clear or abundant signals attributable to other Si-containing compounds. If present, their concentrations are likely low and below the level of unambiguous identification with the current dataset.

**2. Can these methylsiloxanes compounds be resolved and analyzed by other analytical instrumentation, e.g. HPLC-Orbitrap MS? If possible, please discuss the current available method and analytical challenge of methylsiloxanes in the introduction section.**

**Response:** At present, most available analytical techniques are primarily suited to the detection of small molecular methylsiloxanes. The large molecular methylsiloxanes observed in this study have molecular weights that exceed the direct detection range of conventional mass spectrometric approaches. In principle, alternative platforms such as HPLC–Orbitrap MS could be applicable if coupled with a thermal desorption (TD) step. However, without such thermal pretreatment, these large molecular species remain difficult to resolve. Moreover, the identification of methylsiloxanes and their derivatives, particularly the assignment of characteristic and isotopic ion patterns, poses additional analytical challenges. In this respect, the TD-based approach employed here provides a practical advantage by enabling the conversion of large molecular methylsiloxanes into smaller, diagnostic fragments that can be more reliably identified. We have added a brief discussion of these methodological considerations and analytical challenges to the Introduction section.

"Notably, the methylsiloxanes emitted by ships and vehicles include a significant fraction of large molecular methylsiloxanes, which are not directly detectable by conventional mass spectrometry due to their high molecular weights but can be identified following high-temperature thermal depolymerization into smaller fragments." (Page 2, Line 54–56)

**3. It seems the quantification of individual methylsiloxanes was established based on PTR transmission curve. Could any commercial methylsiloxanes standards be used for establishing calibration curve? For instance, I wonder whether this quantification method could be tested or validated for methylsiloxanes standards, e.g. by spiking methylsiloxanes standards with known mass onto the filter? This could examine the reliability of this quantification method.**

**Response:** The quantification of individual methylsiloxanes in this study is based on the PTR transmission curve. During the establishment of this transmission curve, the mixed calibration gas included volatile methylsiloxane standards (D3, D4, and D5) (Holzinger et al., 2019; Worton et al.,

2023). Therefore, the transmission curve is effectively validated for representative methylsiloxanes, although not for the full range of species observed. Direct validation by spiking known masses of methylsiloxane standards onto filters would indeed be valuable; however, such experiments are currently limited by the availability of suitable standards. We acknowledge this limitation and note that additional validation using spiked filter samples will be pursued in future work.

***4. For each PM sample, the chemical profiles of methylsiloxanes were collected at different temperatures. Those produced at lower temperature were assigned to be methylsiloxanes with small molecular weight, while higher temperature was associated with large molecular weight. It seems this desorbing and/or depolymerization process is highly dynamic. I wonder what is the temporal trend or evolution process from the PTR results by increasing the temperature during the experiment? Can the authors give an example specifically? This would help to understand how individual methylsiloxanes and total methylsiloxanes finally being quantified and converted to mass loading on the filter.***

**Response:** Yes, the temporal evolution of methylsiloxanes during the thermal desorption/depolymerization process can be inferred from our PTR results. Figures 2a and 2b effectively illustrate these trends, although the x-axis is labeled by temperature segments rather than time. Each temperature segment corresponds to a 3-minute interval, so the figure can also be interpreted as a time series with 3-minute steps. The data presented in the main text are representative, showing both the total methylsiloxane signal and the contributions from individual small molecular species within each segment. To further clarify this temporal evolution, we have added an example, Fig. S10, in the Supporting Information, which shows the variation of the D3 main peak intensity over time.

[Figure]

**Fig. S10. Temporal variation of the D3 main peak (m/z 223.063) intensity in a single sample.** Raw signals are normalized to represent peak intensity.

**5. Section 2.3 - For D5 (C10H30O5Si5), the first isotope peak (m/z 372) you mentioned actually merges both C13 and Si29 isotopes, and therefore in total accounts for 36% of main peak at m/z 371. Same for the second isotope peak at m/z 373. Am I right?**

**Response:** Yes, the reviewer's interpretation is essentially correct. The first isotope peak of D5 at m/z 372 primarily results from the combined contributions of $^{29}$Si and $^{13}$C isotopes, which together account for approximately 36% of the intensity of the main peak at m/z 371. Minor additional contributions from $^{17}$O and $^{2}$H are also present but are comparatively smaller. Similarly, the second isotope peak at m/z 373 includes contributions from higher-order isotopes, notably $^{30}$Si, following the same underlying isotopic logic.

**6. Section 4.1-4.3 could be merged as one section in Section 3.4 as "atmospheric implication", and the discussion of health and climate relevant impact could be shortened and condensed a bit.**

**Response:** We appreciate the reviewer's suggestion. Sections 4.1–4.3 were separated into individual subsections to present the discussion and implications more clearly, given the substantial amount of content. While these points could conceptually be merged into a single implications section, doing so would make it difficult to convey the results and their interpretation in sufficient detail. We have already substantially condensed this part of the manuscript, with a significant portion of the calculations and analyses moved to the Supporting Information. Further condensation could risk oversimplifying the findings and reducing the clarity of the scientific message. Therefore, we respectfully prefer to retain the current subsection structure and discussion in the main text.

**References**

Holzinger, R., Acton, W. J. F., Bloss, W. J., Breitenlechner, M., Crilley, L. R., Dusanter, S., Gonin, M., Gros, V., Keutsch, F. N., Kiendler-Scharr, A., Kramer, L. J., Krechmer, J. E., Languille, B., Locoge, N., Lopez-Hilfiker, F., Materić, D., Moreno, S., Nemitz, E., Quéléver, L. L. J., Sarda Esteve, R., Sauvage, S., Schallhart, S., Sommariva, R., Tillmann, R., Wedel, S., Worton, D. R., Xu, K., and Zaytsev, A.: Validity and limitations of simple reaction kinetics to calculate concentrations of organic compounds from ion counts in PTR-MS, Atmos. Meas. Tech., 12, 6193–6208, https://doi.org/10.5194/amt-12-6193-2019, 2019.

Worton, D. R., Moreno, S., O'Daly, K., and Holzinger, R.: Development of an International System of Units (SI)-traceable transmission curve reference material to improve the quantitation and comparability of proton-transfer-reaction mass-spectrometry measurements, Atmos. Meas. Tech., 16, 1061–1072, https://doi.org/10.5194/amt-16-1061-2023, 2023.